# The Downregulation of the Liver Lipid Metabolism Induced by Hypothyroidism in Male Mice: Metabolic Flexibility Favors Compensatory Mechanisms in White Adipose Tissue

**DOI:** 10.3390/ijms251910792

**Published:** 2024-10-08

**Authors:** Lamis Chamas, Isabelle Seugnet, Odessa Tanvé, Valérie Enderlin, Marie-Stéphanie Clerget-Froidevaux

**Affiliations:** 1CNRS/MNHN UMR 7221 “Physiologie Moléculaire et Adaptation” Phyma, Department of “Life Adaptations” Muséum National d’Histoire Naturelle 57, Rue Cuvier CP 32, 75231 Paris, CEDEX 05, France; 2Paris-Saclay Institute of Neuroscience (Neuro-PSI), CNRS UMR 9197, Université Paris-Saclay, 91400 Saclay, France; valerie.enderlin@universite-paris-saclay.fr

**Keywords:** hypothyroidism, metabolic flexibility, lipogenesis, white adipose tissue, liver, PTU (propylthiouracil)

## Abstract

In mammals, the maintenance of energy homeostasis relies on complex mechanisms requiring tight synchronization between peripheral organs and the brain. Thyroid hormones (THs), through their pleiotropic actions, play a central role in these regulations. Hypothyroidism, which is characterized by low circulating TH levels, slows down the metabolism, which leads to a reduction in energy expenditure as well as in lipid and glucose metabolism. The objective of this study was to evaluate whether the metabolic deregulations induced by hypothyroidism could be avoided through regulatory mechanisms involved in metabolic flexibility. To this end, the response to induced hypothyroidism was compared in males from two mouse strains, the wild-derived WSB/EiJ mouse strain characterized by a diet-induced obesity (DIO) resistance due to its high metabolic flexibility phenotype and C57BL/6J mice, which are prone to DIO. The results show that propylthiouracil (PTU)-induced hypothyroidism led to metabolic deregulations, particularly a reduction in hepatic lipid synthesis in both strains. Furthermore, in contrast to the C57BL/6J mice, the WSB/EiJ mice were resistant to the metabolic dysregulations induced by hypothyroidism, mainly through enhanced lipid metabolism in their adipose tissue. Indeed, WSB/EiJ mice compensated for the decrease in hepatic lipid synthesis by mobilizing lipid reserves from white adipose tissue. Gene expression analysis revealed that hypothyroidism stimulated the hypothalamic orexigenic circuit in both strains, but there was unchanged melanocortin 4 receptor (*Mc4r*) and leptin receptor (*LepR*) expression in the hypothyroid WSB/EiJ mice strain, which reflects their adaptability to maintain their body weight, in contrast to C57BL/6J mice. Thus, this study showed that WSB/EiJ male mice displayed a resistance to the metabolic dysregulations induced by hypothyroidism through compensatory mechanisms. This highlights the importance of metabolic flexibility in the ability to adapt to disturbed circulating TH levels.

## 1. Introduction

In mammals, the maintenance of energy homeostasis relies on complex mechanisms requiring tight synchronization between peripheral organs and the brain. It involves both the endocrine system and central nervous system (CNS). The hypothalamus plays an essential role in the central control of energy balance, especially as a nutrient sensor modulating food intake and energy expenditure according to the metabolic status [1]. It orchestrates a dynamic crosstalk between its different nuclei and the peripheral metabolic tissues. The first-order neurons present in the arcuate nucleus (ARC) integrate peripheral signals such as hormones (leptin, insulin, etc.) and nutrients (lipids and glucose) to monitor the metabolic status of the entire organism [1,2]. In response, potent outputs are conveyed through orexigenic (neuropeptide Y (NPY)/Agouti-related peptide (AGRP)) or anorexigenic (pro-opiomelanocortin (POMC)/cocaine- and amphetamine-regulated transcript (CART)) neurons to the second-order neurons located in other nuclei of the hypothalamus, such as the paraventricular nucleus (PVN), which will eventually restore energy homeostasis, ensuring weight maintenance by modulating energy expenditure and food intake [3]. One of the main actors in maintaining this energy balance is the hypothalamus–pituitary–thyroid axis. This axis controls the synthesis of thyroid hormones (THs), which, as one of their pleiotropic actions, act as key regulators of energy balance [4]. Thyroxine (T4) and the transcriptionally active triiodothyronine (T3) ensure metabolic homeostasis of the whole body mainly through the transcriptional activity of their nuclear receptors (thyroid receptors, TRs). They control all aspects of metabolism, acting both centrally and in peripheral metabolic organs (particularly the liver and adipose tissues) [5]. THs act on lipid and carbohydrate metabolism, as well as on metabolic cellular mechanisms, such as mitochondrial activity and thermogenesis, and on food intake and energy expenditure [4,6]. Due to the importance of THs in metabolic control, any disruption of TH homeostasis could cause metabolic disorders [7]. Hypothyroidism, which is characterized by low circulating TH levels, slows down the metabolism. This can lead to a reduction in energy expenditure, as well as in lipid and glucose metabolism, and to insulin resistance [4,8,9]. Interestingly, comparable metabolic changes are also present in patients with thyroid hormone resistance (THR), a syndrome in which patients are insensitive to T3 due to mutations in the hormone-binding domain of certain TRs. These data show that the effects of THs on metabolic regulation could also be directly impacted by the deregulation of T3-responsive genes involved in metabolic homeostasis [6].

Other factors can, however, disrupt energy homeostasis. Foods that are rich in fat and sugar can also lead to metabolic dysregulations and the development of metabolic disorders, the most common being obesity and insulin resistance [10,11,12]. Research on these metabolic disorders commonly uses the diet-induced obesity (DIO) model, which consists of feeding mice a high-fat diet (HFD) to induce obesity and metabolic syndrome. Interestingly, some mouse strains are resistant to DIO, such as the wild-derived WSB/EiJ strain. This could thus be a particularly useful model to unravel the pathways involved in metabolic homeostasis. Indeed, we and others have previously shown that these mice have an extraordinary ability to maintain metabolic homeostasis with strong DIO resistance [13,14]: they do not gain weight after 3 days or 8 weeks of an HFD and consequently do not show any neuroinflammatory responses to an HFD in the hypothalamus region [14]. This is known to be associated with obesity for a long time [12]. In contrast, C57BL/6J mice are prone to obesity in response to an HFD and develop hypothalamic neuroinflammation. Our previous results suggested that the resistance of WSB/EiJ mice could be, at least partially, explained by differences in mitochondrial reactivity and the differential expression of genes involved in inflammatory and mitochondria pathways in response to an HFD, unlike C57BL/6J mice [14]. In addition, in a previous study [15], we showed that in the hippocampus of WSB/EiJ mice, the TRs were fully functional transcriptional factors as reflected by the expression of TRs and THs target genes that were regulated by local T3 levels. Finally, they displayed lower circulating T4 levels (around 3 µg/dL) compared to most other mouse strains, including the C57BL/6J strain. However, it remained in the euthyroid reference range for mice [2.6–9 µg/dL] that was established based on 29 non-pathological mouse strains (Donahue 1 study https://phenome.jax.org/measures/11531 (accessed on 7 September 2024)). This lower circulating T4 level, associated with the DIO resistance, constitutes an interesting enigmatic paradox making WSB/EiJ mice a relevant model for metabolic studies.

Thus, regarding the role played by THs in the control of metabolism, the aim of this article was, first, to evaluate if WSB/EIJ mice were also able to adapt to the metabolic imbalance induced by hypothyroidism contrary to C57BL/6J mice. Secondly, if true, the mechanisms involved in this adaptation were to be elucidated, which would help to identify pathways which are involved in metabolic flexibility and could thus be targeted from a therapeutic perspective.

For this purpose, a transient period of hypothyroidism was induced in adult C57BL/6J and WSB/EiJ male mice by administrating a commonly used antithyroid molecule, the propylthiouracil (PTU), for seven weeks [15]. The consequences of PTU-induced hypothyroidism on peripheral and central metabolism were explored. This was conducted by following weight changes, the expression of genes involved in the hypothalamic control of food intake and energy expenditure, markers of lipid metabolism, as well as the expression of genes involved in metabolic pathways in liver and white adipose tissue. As metabolic unbalance is known to induce hypothalamic inflammation [14,16], the inflammatory status of the hypothalamus was also verified by immunohistochemistry. This study showed that 7 weeks of PTU induced an alteration in lipid metabolism in both strains, particularly a reduction in hepatic lipid synthesis. Furthermore, contrary to the C57BL/6J mice, the WSB/EiJ mice were resistant to the metabolic dysregulations induced by hypothyroidism, mainly by limiting hepatic ß oxidation and stimulating lipid mobilization from eWAT through an enhanced lipid metabolism in adipose tissue.

## 2. Results

### 2.1. Hypothyroidism Consequences on Metabolism and Energy Balance

#### 2.1.1. Body Weight, Food Intake, and Body Fat Mass

The consequences of hypothyroidism were investigated on peripheral and central metabolism between the two strains. The effect of PTU treatment on the body weight of both mouse strains was evaluated, particularly the variation in the weight during the treatment (Figure 1A). An increase in the weight of euthyroid compared to hypothyroid groups was observed, starting from the 1st week of treatment (0.0001 ≤ *p* ≤ 0.001): the body weight of euthyroid C57BL/6J and WSB/EiJ mice significantly increased in a similar way throughout the treatment, reflecting the growth period of young adult mice (0.0001 ≤ *p* ≤ 0.05). In contrast, the weight gain in hypothyroid C57BL/6J mice was inhibited by PTU treatment, which was revealed by the significant decrease in body weight throughout the weeks of treatment. The weight decrease reached 6% of the initial weight at the 5th week of treatment (0.0001 ≤ *p* ≤ 0.05). In contrast, the body weight of hypothyroid WSB/EiJ mice was maintained throughout the treatment (*p* > 0.05), although their body weight was lower than their control group’s weight which increased over the weeks of treatment (0.0001 ≤ *p* ≤ 0.001). Thus, the PTU treatment induced different effects between the hypothyroid groups: C57BL/6J mice lost weight, whereas WSB/EiJ maintained their body weight during the treatment.

In order to verify if the difference in body weight gain observed in hypothyroid mice was due to a lower food consumption induced by PTU treatment, the food intake of each group was measured (Figure 1B). Interestingly, a difference in food intake was observed between the two euthyroid groups according to the strain (*p* < 0.0001). Euthyroid WSB/EiJ mice have a higher food intake than C57BL/6J mice despite their lean phenotype, reflecting the efficient metabolic character of this strain. It is worth noting that both euthyroid C57BL/6J and WSB/EiJ mice maintained their food intake throughout the weeks of treatment (*p* > 0.05) except for the fact that the food intake of euthyroid C57BL/6J mice was reduced during the very last week of the treatment (probably because mice reached a stable body weight, *p* < 0.05). However, PTU treatment significantly reduced the food intake in both hypothyroid C57BL/6J and WSB/EiJ mice compared to their euthyroid control groups (*p* < 0.05). For both hypothyroid groups, the food intake was comparably lower than the euthyroid mice from the first week of treatment (*p* < 0.05) and then was maintained throughout the treatment (*p* > 0.05). Therefore, the significant weight loss observed in hypothyroid C57BL/6J mice (and not in hypothyroid WSB/EiJ mice) was not due to a lower food intake, as both strains maintained their food consumption during the treatment.

Furthermore, the different weight was not due to growth impairment as assessed by femur length measurements, which revealed no differences between the groups (Figure 1C). Although euthyroid WSB/EiJ mice consumed more food than C57BL/6J mice related to their weight, their eWAT weight was considerably (two times) lower than C57BL/6J mice (*p* = 0.002; Figure 1D), highlighting the low-fat mass in WSB/EiJ mice at the basal state. In response to PTU treatment, eWAT weight was unchanged in C57BL/6J mice (*p* > 0.05), whereas the eWAT weight was decreased in WSB/EiJ mice (*p* = 0.004). Since leptin is known to be proportional to body fat mass (Figure 1E), circulating leptin levels were accordingly lower in euthyroid WSB/EiJ mice compared to C57BL/6J mice (strain effect *p* = 0.0003), reflecting once again the lean phenotype of the WSB/EiJ strain. In response to PTU treatment, circulating leptin levels were decreased in C57BL/6J mice (*p* = 0.0003), whereas levels were maintained in WSB/EiJ mice (*p* > 0.05). This is consistent with the weight loss of C57BL/6J and the weight maintenance of WSB/EiJ mice (strain × treatment interaction *p* < 0.00001). These results showed that PTU treatment altered the body weight and the fat mass in different ways between both strains.

#### 2.1.2. Hypothalamic Regulation of Energy Balance

Due to the observed effects of PTU treatment on body weight and food intake, the effect of hypothyroidism was further investigated on the hypothalamic expression of genes involved in the control of these parameters (Figure 2). Particularly, the expressions of orexigenic (*Agrp*) and anorexigenic factors (*Pomc*, *Mc4R*, *LepR*) were evaluated. The expression of *Agrp* was upregulated by 138% and 83% in hypothyroid C57BL/6J (*p* < 0.05) and WSB/EiJ mice, respectively, compared to their euthyroid group. However, the difference was not statistically significant for WSB/EiJ mice due to the high variability of this group (*p* > 0.05; Figure 2A). In contrast, the expression of anorexigenic factors was downregulated in hypothyroid C57BL/6J mice after PTU treatment compared to controls (*p* < 0.05; Figure 2B,D, see Appendix A). For WSB/EiJ mice, the expression of *Mc4r* and *Lepr* was maintained (*p* > 0.05), whereas *Pomc* expression was decreased in hypothyroid mice compared to their control group (treatment effect *p* = 0.007). Taken together, these results showed that hypothyroidism stimulates the hypothalamic orexigenic circuit in both strains. However, unchanged *Mc4r* and *LepR* expression in hypothyroid WSB/EiJ mice strain reflected their adaptability to maintain their body weight contrary to C57BL/6J mice.

#### 2.1.3. Circulating Lipids

Circulating lipid profiles were assessed. Both C57BL/6J and WSB/EiJ mouse strains exhibited an increase in cholesterol levels in response to PTU (*p* < 0.001; Figure 3A), which was even more pronounced in WSB/EiJ mice (by 29% compared to hypothyroid C5BL/6J mice; strain × treatment interaction *p* = 0.03). Low-density lipoprotein (LDL) levels, which carried cholesterol into the blood, followed the same profile as cholesterol levels for both hypothyroid strains (treatment effect *p* = 0.0001; strain × treatment interaction *p* = 0.01; Figure 3B). Similarly, high-density lipoprotein (HDL) levels, which export cholesterol from blood to the liver for clearance, increased in both hypothyroid mouse strains (treatment effect *p* = 0.0001; Figure 3C). Therefore, both mouse strains exhibited hypercholesterolemia in response to hypothyroidism. In contrast, triglycerides (TGs) and non-esterified fatty acid (NEFA) levels were differentially impacted by hypothyroidism between both strains (strain × treatment interaction *p* = 0.02): they decreased in hypothyroid C57BL/6J mice (*p* < 0.01), whereas levels remained unchanged in hypothyroid WSB/EiJ mice (*p* > 0.05; Figure 3D,E). These results were in line with the difference in body weight observed between the two hypothyroid strains. This reduction in TGs and NEFA levels in hypothyroid C57BL/6J mice, but not in WSB/EiJ mice, suggested an alteration in lipid content only in hypothyroid C57BL/6J mice.

#### 2.1.4. Hepatic and Adipose Lipid Metabolism

In order to better understand the differential lipid responses to hypothyroidism between the two strains, the potential effect of PTU treatment on the peripheral lipid metabolism was investigated. Thus, the expression of lipid-related genes in the liver, the main tissue in which lipid synthesis occurs, was quantified by RT-qPCR (Figure 4). The expression of lipogenic genes (*Fasn*, *Acacα*), key actors of the lipogenesis pathway by which TGs are synthetized, were downregulated by 2-fold in both hypothyroid mouse strains compared to the untreated mice (Figure 4A,B) (treatment effect *p* = 0.0003, see Appendix A).

Furthermore, the transcription factor *Chrebp*, which is involved in the regulation of this pathway, was also downregulated in both hypothyroid strains compared to their euthyroid group (treatment effect, *p* = 0.0001; Figure 4C). The expression of *Pparα*, implicated in hepatic lipid metabolism and utilization, was downregulated in both hypothyroid strains compared to their euthyroid control (treatment effect *p* = 0.0001; Figure 4D). However, the expression of *Ppargc1α*, known as a co-activator of *Pparα* and essential for fatty acid oxidation, was upregulated in C57BL/6J mice in response to PTU treatment (*p* < 0.01). However, this remained unaffected in WSB/EiJ mice (strain × treatment interaction *p* = 0.04; Figure 4E). Another key actor of fatty acid oxidation, *Fgf21*, was also upregulated in hypothyroid C57BL/6J mice but slightly downregulated in WSB/EiJ mice (0.01 < *p* < 0.05; Figure 4F). Taken together, these results showed that PTU treatment impaired the hepatic TGs synthesis in both strains, whereas FFA oxidation was differentially affected by hypothyroidism depending on strains.

Next, the mechanisms by which hypothyroid WSB/EiJ mice were able to maintain their circulating lipid contents were analyzed, as TGs synthesis was reduced in the liver. A potential source of lipids is lipolysis, which catalyzes TGs (stored in the form of lipid droplets) into free-fatty acid (FFA) by the PNPLA2 enzyme in the eWAT, which is another site of lipogenesis. In response to PTU treatment, the mRNA expression of this key enzyme was unaffected in C57BL/6J mice (*p* > 0.05), whereas it was upregulated in hypothyroid WSB/EiJ mice (by 110% compared to the euthyroid group, *p* > 0.05; Figure 5A). However, the difference observed in hypothyroid WSB/EiJ mice did not reach significance due to the high variability of this group. In addition, the expression of *Ppargc1a*, the regulator of FFA oxidation in the eWAT, was quantified (Figure 5B). *Ppargc1α* mRNA expression was downregulated in both hypothyroid groups (*p* < 0.01) and more importantly (2-fold compared to euthyroid) in hypothyroid C57BL/6J mice compared to WSB/EiJ mice (strain × treatment *p* = 0.02). Finally, the expression of lipid-related genes in the eWAT was quantified. The expression of *Fasn* and *Acacα* remained unchanged in hypothyroid C57BL/6J mice (*p* > 0.05), whereas it was upregulated by almost 3-fold in hypothyroid WSB/EiJ mice (strain × treatment interaction *p* ≤ 0.01; Figure 5C,D), suggesting an enhanced lipogenesis only in WSB/EiJ mice in response to PTU treatment.

Although lipogenesis is regulated by the transcription factor *Pparγ*, its expression was unchanged in both hypothyroid strains (*p* > 0.05; Figure 5E).

Thus, globally, lipid metabolism in eWAT was improved in WSB/EiJ mice in response to PTU treatment and not in C57BL/6J mice, which could explain the sustained circulating lipid levels in the WSB/EiJ strain. Consequently, C57BL/6J mice were more affected by hypothyroidism than WSB/EiJ mice, which developed mechanisms to compensate for lipid deficiency.

### 2.2. Hypothyroidism Consequences on Peripheral and Hypothalamic Inflammation

#### 2.2.1. Peripheral Inflammation

Because lipid metabolism was dysregulated mainly in hypothyroid C57BL/6J mice, it was questioned whether an inflammatory response was developed within the two mouse strains (Figure 6). The expression of pro-inflammatory (*Tnfα*, *Il6*, *Il1β*) and anti-inflammatory (Il10) genes were first evaluated in the eWAT organ, which is actively producing and secreting a plethora of cytokines and hormones ensuring homeostasis [17] (Figure 6A). The expression of these inflammatory genes was overall downregulated in hypothyroid C57BL/6J mice (*p* < 0.05, except for *Il6*), whereas it was unchanged in WSB/EiJ mice (*p* > 0.05, except for *Il10*) compared to the untreated mice.

These results were confirmed by circulating cytokine measurements that showed similar profile patterns (Figure 6B). The cytokine levels were overall lower in WSB/EiJ mice than in C57BL/6J mice according to the strain (*p* < 0.05).

In response to PTU treatment, serum IL1β and IL10 levels were decreased in C57BL/6J mice (by 77% and 27%, respectively compared to their euthyroid group; *p* < 0.05). Inversely, IL1β levels were increased in WSB/EiJ mice (by 110% compared to their control group, *p* < 0.05), whereas IL10 levels were unaffected in response to hypothyroidism (*p* < 0.05). IFNγ and IL-6 levels remained unchanged in both mouse strains in response to PTU treatment (*p* > 0.05). Overall, PTU treatment induced differential peripheral inflammatory responses between strains that were reduced in C57BL/6J mice and maintained in WSB/EiJ mice.

#### 2.2.2. Hypothalamic Inflammation

Central inflammation was also investigated by quantifying astrocytes and microglia populations as inflammatory markers. Hypothalamic glial cells were first checked in the ARC (Figure 7). In agreement with peripheral inflammatory markers, the density of IBA1 + microglia was slightly decreased in hypothyroid C57BL/6J mice (by 14% compared to their euthyroid control, *p* < 0.05), whereas it was maintained in hypothyroid WSB/EiJ mice (strain × treatment interaction *p* = 0.04) (Figure 7A,B).

The density of GFAP+ astrocytes was unaffected by PTU treatment in both strains (treatment effect *p* > 0.05) (Figure 7C,D). However, it was overall lower in WSB/EiJ mice compared to C57BL/6J mice, indicating a lower level of inflammatory markers in this strain. Therefore, these results showed that inflammatory markers were reduced in C57BL/6J mice, whereas they remained unchanged in WSB/EiJ mice in response to hypothyroidism, which is consistent with the reduction in peripheral lipid metabolism in the C57BL/6J strain.

## 3. Discussion

HPT axis is a key actor in the regulation of energy balance, and hypothyroidism is well described to cause metabolic dysregulations in rodents and humans [4,6,7] but leading to opposite effects on weight between rodents and humans, in whom hypothyroidism is usually associated with weight gain. In our study, in response to hypothyroidism, the metabolic dysregulations were first manifested by weight loss in both mouse strains compared to their euthyroid control. This was similarly observed in hypothyroid rodents treated with PTU or MMI, beginning from 2 weeks of treatment [18,19,20]. This weight loss induced by hypothyroidism was likely partially due to an alteration of food intake mechanisms as both strains reduced their consumption when treated with PTU compared to euthyroid mice. This was also observed in hypothyroid rodents associated with a weight loss [19,20]. Conversely, an increase in food intake and weight gain was observed in hyperthyroid mice treated with T4 for 9 weeks [21]. In response to PTU treatment, a differential body weight variation was observed between hypothyroid mice of both strains that was independent of the food intake, as it was maintained throughout the treatment. More specifically, the hypothyroid WSB/EiJ mice maintained their body weight throughout the treatment, whereas hypothyroid C57BL/6J mice showed a weight loss of 6% that was pronounced at week five of the treatment. The differential body weight between the two strains was reflected by leptin concentrations, which are known to be secreted by the white adipose tissue (WAT) in proportion to body fat [22]. The reduced serum leptin levels in hypothyroid C57BL/6J mice reflected their reduced body fat, whereas these levels were maintained in WSB/EiJ mice accordingly to their stable body weight in response to treatment. The hypothalamus is the key regulator of energy balance and body weight maintenance [2,23]. It involves the coordination of two main hypothalamic neuronal circuits: orexigenic and anorexigenic circuits, particularly AGRP/NPY and POMC/MC4R neurons, respectively [1,2]. Moreover, metabolic signals, such as leptin, orchestrate the regulation of these neuronal circuits in order to modulate the food intake. In response to hypothyroidism, mRNA levels showed a stimulation of the orexigenic neurons (*Agrp*) and an inhibition of the anorexigenic neurons (*Pomc*), as well as a decrease in *LepR* expression, suggesting a coordinated response observed in both strains in order to increase food intake to restore the stocks. However, in the study by Herwig et al. [20], in hypothyroid rats, *Agrp* was decreased by induced hypothyroidism, which was concomitant with weight loss and food intake diminution. This difference in orexigenic expression could be explained by the fact that rats were less hypothyroid than the mice of the present study, which was due to MMI treatment instead of PTU to induce hypothyroidism. Another factor could be that the treatment was shorter (25 days instead of 48 days). Indeed, as already hypothesized above, increased *Agrp* expression could be a reaction against long-term weight loss and the expression of a resistance to orexigenic signals that is induced by prolonged hypothyroidism. Both studies confirmed that inducing hypothyroidism in rodents impaired the hypothalamic control of food intake, leading to weight loss. In the current study, the activation of orexigenic pathways in response to hypothyroidism could explain the maintenance of food consumption in both hypothyroid mice strains throughout the treatment. However, their food intake remained lower than that of euthyroid mice, suggesting a clear effect of hypothyroidism on the control of appetite. It is interesting to note that, as already observed in a former study from our lab, basal levels of *Agrp* were higher in WSB/EiJ than in C57BL/6J, which confirmed the permanent negative energy balance of the WSB/EiJ strain [14]. Thus, the stimulation of the orexigenic neurons could be associated with the decrease in leptin levels in hypothyroid C57BL/6J as their weight was reduced. Therefore, the classical leptin action was observed as a negative feedback loop in hypothyroid C57BL/6J mice to maintain the energy balance and favor the recovery of weight loss. The combination of these responses has been well demonstrated in diet-induced weight loss conditions such as starvation and caloric restriction [24,25]. However, as mentioned above, leptin levels were maintained in hypothyroid WSB/EiJ mice, which was consistent with stable body weight. This presumes that the stimulation of orexigenic circuits in WSB/EiJ mice is not dependent on leptin signaling, as observed in C57BL/6J mice, and underlies the involvement of other homeostatic mechanisms such as the melanocortin pathway. The hypothalamic melanocortinergic pathway is known to be involved in the regulation of appetite and weight maintenance [26]. It is triggered when POMC is activated and AGRP/NPY is inhibited, thus mediating anorexigenic signals through melanocortin receptors (MC4R/MC3R) [27,28]. As the orexigenic circuit was stimulated in hypothyroid mice, the expression of the *Mc4r* gene was reduced accordingly in hypothyroid C57BL/6J mice as it follows the reduction in the *Pomc* gene. However, mRNA levels of *Mc4r* remained unchanged in hypothyroid WSB/EiJ mice despite the downregulation of the *Pomc* gene. Therefore, the melanocortin pathway seems to be reduced in C57BL/6J mice, whereas it is maintained in the WSB/EiJ strain, explaining the weight difference between both hypothyroid strains. Moreover, it has been reported that the expression of the *Mc4r* gene is negatively regulated by TH, since its expression is upregulated in hypothyroid mice [19]. This contradicts our results as the mRNA level of the *Mc4r* gene did not increase in hypothyroid mice. This could be due to the difference in the duration of the treatment that was applied: hypothyroidism was induced for a short period of time (13 days) in our previous study contrary to the duration of this experiment (7 weeks). This suggests that the homeostatic regulation of the *Mc4r* gene by thyroid status may be disrupted by prolonged and severe hypothyroidism, compromising the activation of anorexigenic mechanisms (as observed in C57BL/6J mice). Consequently, it was hypothesized that in WSB/EiJ mice, the melanocortin pathway may be regulated by other mechanisms in response to hypothyroidism, allowing mice to maintain their body weight, contrary to C57BL/6J mice. These differential responses between the two strains highlight the resistant phenotype of WSB/EiJ mice to hypothyroidism that relies on their ability to maintain energy homeostasis through adaptive mechanisms.

Numerous studies have reported a decrease in lipid levels in response to hypothyroidism. In mice treated with PTU for 7 weeks, a decrease in serum TG and FFA levels was observed [29]; a decrease in plasma TGs concentrations associated with a reduction in fat mass were also reported in rats treated with MMI for 22 days [20]. In our study, in hypothyroid C57BL/6J mice, FFA and TG levels were decreased, whereas they remained unchanged in hypothyroid WSB/EiJ mice, revealing a differential hypothyroidism effect on the lipid machinery between both strains. These lipid concentrations are concordant with the weight differences as well as the leptin levels observed between hypothyroid mouse strains.

Taken together, these data confirmed that PTU treatment induced an alteration in lipid metabolism, particularly in C57BL/6J mice. In contrast, WSB/EiJ mice showed a phenotype of resistance to lipid alteration induced by hypothyroidism. THs are critical actors of the peripheral metabolism by exerting direct effects on metabolically active organs such as the liver and WAT [4,30]. It has been extensively described in humans and rodents that hypothyroidism decreased the carbohydrate and lipid metabolism, leading to an insulin-resistance state [9,31,32]. FFAs and TGs are the main lipid components produced particularly in the liver. THs are a well-known activator of de novo hepatic lipogenesis (DNL) by which the FFAs are synthetized, as they directly stimulate the transcription of several key genes such as *Fasn* and *Acac*α. THs control the expression of transcription factors involved in the DNL such as *Chrebp* [33,34]. After their synthesis, FFAs are esterified, accumulated as TGs and released in blood circulation. In this present study, hypothyroidism induced an impairment of the lipid metabolism, which was first reflected by the decrease in hepatic lipid synthesis. This was revealed by the downregulation of the lipogenic genes’ expression, mainly *Fasn*, *Acacα* and *Cherbp*, in the liver of both strains, suggesting a reduced hepatic DNL and thus lipid synthesis. Similar results were previously observed in hypothyroid rodents [35,36]. Also, an increase in *Ppargc1α* and *Fgf21* expression, which are TH target genes involved in hepatic ß-oxidation [33], was observed in the liver of the C57BL/6J mice but not in WSB/EiJ mice. This is in accordance with the triglycerides and NEFA circulating levels which are decreased only in C57BL/6J mice. Moreover, eWAT DNL was differentially altered between both hypothyroid strains in response to the hepatic lipid deficiency. Adipocyte DNL is an important source of endogenous lipids and plays crucial roles in maintaining systemic metabolic homeostasis. The biosynthesis of TGs in the eWAT results from FFAs synthesis by DNL in case of fat excess. Inversely, lipolysis is the catabolic process leading to the breakdown of TGs by adipose triglyceride lipases such as PNPLA2 to FFAs that can be released and delivered to the liver when energy fuel is required [37,38]. Both adipocyte lipogenesis and lipolysis are regulated by THs [39,40]. In C57BL/6J mice, hypothyroidism had no effect on either adipose lipogenesis nor lipolysis, as the expression of respective genes (*Fasn*, *Acacα*, *Pnpla2*) remained unchanged. In contrast, adipose lipogenic and lipolytic genes were considerably upregulated in hypothyroid WSB/EiJ mice, suggesting an enhanced lipid mobilization in eWAT in response to the reduced hepatic lipid machinery.

Thus, the results of this study demonstrated that hypothyroidism reduced hepatic lipid metabolism in both strains but was rescued only in WSB/EiJ mice by increasing lipid availability from adipocytes. This could explain the reduced circulating lipid levels (triglycerides and NEFA) in hypothyroid C57BL/6J mice, whereas they were maintained in WSB/EiJ mice. Furthermore, eWAT weight was decreased only in hypothyroid WSB/EiJ mice, which could result from hydrolyzed fat occurring in this tissue, highlighting once again the metabolic flexibility of WSB/EiJ mice.

Combining these results indicates that WSB/EiJ mice could compensate for the reduced lipid hepatic biosynthesis induced by hypothyroidism in both strains by limiting hepatic ß oxidation and stimulating lipid mobilization from eWAT.

In metabolic dysregulations such as obesity, a low-grade inflammatory response is triggered by eWAT, which releases cytokines and chemokines [12,41]. One of the aims of this study was to understand if metabolic dysregulations induced by hypothyroidism promote peripheral inflammation. The inflammatory response was overall reduced in C57BL/6J mice, whereas it was maintained in WSB/EiJ mice in response to PTU treatment. Pro-inflammatory cytokine transcripts, such as *Tnf*α and *Il1b*, were downregulated in the eWAT of hypothyroid C57BL/6J, whereas they remained unaltered in WSB/EiJ mice, which was in parallel with circulated cytokine levels. Inversely to the consequences of lipid overload [12], it was speculated that lipid deficiency and the weight loss occurring in hypothyroid C57BL/6J mice could partially explain the repression of the inflammatory responses in this strain. The absence of inflammatory response could indeed be explained by the reduction in lipid level known to stimulate pro-inflammatory cytokines [42]. Corroborating other findings, the reduction in body fat is positively correlated with the immune function in many species, including humans [43,44]. However, the maintenance of inflammatory status in WSB/EiJ mice could result from their compensatory mechanisms deployed to overcome THs deficiency, such as weight maintenance. This highlights here again the efficiency of this strain to sustain homeostasis despite peripheral hypothyroidism.

In the brain, THs are key regulators of glial cells, particularly astrocytes and microglia, which are crucial for neuronal protection and brain homeostasis [45]. Moreover, glial cells are considered as brain immune cells, particularly the microglia, since they promote inflammatory processes in inflammatory-related diseases such as obesity [12]. In the hypothalamus, these two glial cell populations are highly sensitive to metabolic deregulations and act as peripheral metabolic sensors mainly in the mediobasal hypothalamus (including the ARC) considered as an interface between the CNS and the periphery [14,46,47]. In the current study, hypothyroidism altered microglia density as it was reduced only in the ARC of C57BL/6J mice (in the range of 14%), whereas the astrocytes density was unaffected in both strains. The decrease in microglia population was also reported in hypothyroid neonatal rat, since THs control microglia development and maturation [48]. It could be hypothesized that the decrease in hypothalamic microglia density in hypothyroid C57BL/6J mice could result from the reduction in peripheral inflammation in response to the declined lipid metabolism induced by hypothyroidism.

## 4. Materials and Methods

### 4.1. Animals and Treatment

Animals were housed individually under a 12:12 light dark cycle (07:00–19:00), maintained at 23 °C, with food and drinking water provided ad libitium. Wild-type C57BL/6J and WSB/EiJ breeder mice were purchased from Charles River (L’Arbresle, France) and Jackson Laboratories (Maine, Bar Harbor, ME, USA), respectively. Hypothyroidism was induced in 8-week-old male mice by given an iodine-deficient diet supplemented with 0.15% propylthiouracil (PTU, Envigo Teklad, Madison, WI, USA) for 7 weeks [15], whereas euthyroid control (CTRL) mice were fed with standard chow diet. The anti-thyroid molecule PTU blocks the activity of thyroid peroxidases that catalyze iodination of thyroglobulin and are essential for thyroid hormone synthesis. Moreover, this molecule also inhibits deiodinase I (Dio1), which produces T3 by the deiodination of T4 in peripheral tissues, such as blood, liver, or kidney [49,50]. Here, thyroid status was assessed by an ELISA kit (Labor Diagnostika Nord (LDN), Nordhorn, Germany) [15]. Peripheral hypothyroidism was confirmed by the expression of Dio1, which is a key marker of peripheral hypothyroidism [51]: Dio1 was drastically downregulated in both hypothyroid strains compared to their euthyroid group (*p* < 0.05; Appendix A). Body and food weights were monitored twice a week during the treatment. All procedures were conducted according to the principles and procedures in Guidelines for Care and Use of Laboratory Animals and were validated by the MNHN ethical comity for animal experimentations (68.096).

### 4.2. Blood and Tissue Sample Collection

At the end of the treatment, retro-orbital blood was collected in the morning, before their euthanasia by decapitation, to measure circulating metabolic parameters. Trunk blood and tissue samples were also collected at this time. Regarding hypothalamus collection, it was dissected along the following boundaries, using curved forceps: laterally 2 mm either side of the third ventricle, from the optic chiasm to the posterior border of the mammillary bodies, and the thalamus dorsally. Blood samples were centrifuged (15 min, 3000× *g*, room temperature), and serum supernatants were stored at −20 °C until analysis. Ependymal white adipose tissue (eWAT) samples were weighed before collection. All tissues were frozen in liquid nitrogen and stored at −80 °C until further analysis.

### 4.3. Circulating Metabolic Parameters and Cytokines Profile

Leptin concentrations were measured using an ELISA kit (EZML-82K, Millipore, Sigma-Aldrich Chimie S.a.r.l., Saint-Quentin Fallavier, France) according to the manufacturer’s instruction. Circulating cytokine and lipid profiles were assayed as described previously [14].

### 4.4. Reverse-Transcription qPCR

The total RNA from hypothalamus, liver and eWAT samples (*n* = 6 per group) were extracted using RNABle lysis reagent (Eurobio, Les Ulis, France) and the RNeasy^®®^ Mini Kit (Qiagen, Hilden, Germany) according to the manufacturer’s protocol. All RNA samples were quantified using a Qubit 2.0 Fluorometer (Invitrogen Life Technologies, Carlsbad, CA, USA), and RNA integrity was evaluated using an Agilent 2100 Bioanalyzer. Complementary DNA (cDNA) synthesis was performed using a Reverse Transcription Master Mix from Fluidigm^®^ according to the manufacturer’s protocol with random primers. Real-time quantitative PCR was carried out with a QuantStudio 6 Flex Real-Time PCR System (Applied Biosystems, Thermo Fisher Scientific, Waltham, MA, USA) using a TaqMan Universal PCR master mix (Applied Biosystems) and pre-designed TaqMan probes (TaqMan Gene Expression Assays, Applied Biosystems) for all of the genes listed in Appendix A. The RT-qPCR reactions for each sample were conducted in duplicate, and direct detection of the PCR product was monitored by measuring the increase in fluorescence generated by the TaqMan probe. The qRT-PCR data were analyzed using QuantStudio™ Real-Time PCR Software (version 1.3, Thermo Fisher Scientific, Waltham, MA, USA) and ExpressionSuite software (version 1.0-4, Thermo Fisher Scientific, Waltham, MA, USA). For each tissue, three housekeeping genes were selected based on the method of Vandesompele et al. [52] and using the SlqPCR package (version 1.42.0). A custom R tool was constructed to measure relative gene expression levels according to the ΔΔCT method [53] and to perform non-parametric statistical tests (two-way ANOVA with permutation) as described previously [14]. Graphical representations (boxplot whiskers) were performed using FC values (fold-changes compared to C57BL/6J CTRL) on GraphPad Prism (GraphPad Software Inc., San Diego, CA, USA; version 8.0-2).

### 4.5. Immunohistochemistry

Hypothalamic immunohistochemistry (*n* = 3–4 per group) was assayed as described previously [15]. Hypothalamic sections were incubated with the following primary antibodies: rabbit anti-IBA1 (1/750, Wako Chemicals, Sobioda, Montbonnot-Saint-Martin, France) or chicken anti-GFAP (1/300, Abcam, Paris, France), and secondary antibodies: donkey anti-rabbit (Alexa Fluor 488, 1/500, Invitrogen, Villebon-sur-Yvette, France) and donkey anti-chicken (Alexa 594 nm, 1/500, Invitrogen). Brain sections were observed under a TCS-SP5 Leica confocal microscope using ×400 magnification. Acquisitions were obtained using a max intensity Z projection of 30 μm thick z-stacks (a minimum of 20 images with a z-step of 1.01 µm) for each hypothalamic arcuate nucleus (ARC) region of interest (ROI; left and right side). Quantifications of microglia and astrocytes were performed with ImageJ software (v1.52p, National Institutes of Health, Stapleton, NY, USA) using the cell counter plugin. Results of GFAP and IBA1 densities are presented as the number of immuno-positive cells per mm^2^ for each ROI.

### 4.6. Statistical Analyses

Statistical analyses for body weight and food consumption were carried out using GraphPad Prism (Boston, MA, USA). Variations between strains (C57BL/6J; WSB/EiJ) during the 7 weeks of treatment (CTRL; PTU) were analyzed with the two-way repeated measures ANOVA test followed by Tukey’s multiple-comparisons test. For the rest of the experiments, a non-parametric two-way ANOVA test with permutation was performed using Rscript as described previously (see above). When effect “treatment” and “strain” were only achieved, a non-parametric one-way ANOVA test with permutation was applied in order to see in which strain the effect of treatment was driven. Dixon’s Q-test was used for the identification and rejection of outliers. Differences were considered significant for a *p*-value ≤ 0.05. Unless specifically mentioned, all data are represented as medians using boxplot and min–max whiskers on GraphPad.

## 5. Conclusions

In this study, 7 weeks of PTU treatment induced an alteration of lipid metabolism in both strains, particularly a reduction in hepatic lipid synthesis. However, only WSB/EiJ mice restored their lipid levels by mobilizing lipid machinery in white adipocytes, thus maintaining their body weight (Figure 8). The WSB/EiJ mice displayed a phenotype of resistance to metabolic dysregulations induced by hypothyroidism thanks to compensatory mechanisms occurring in eWAT, which is complementary to their resistance to HFD-induced obesity. These results confirm their adaptive capacities to maintain metabolic homeostasis, namely, their high metabolic flexibility, despite serum hypothyroidism. Furthermore, our data highlight the importance of metabolic flexibility in the ability to adapt to the disturbance in circulating TH levels.

Finally, several studies in rodents as well as in humans suggest that disrupted metabolic homeostasis could be associated with neurodegenerative lesions through the potential activation of neuroinflammatory pathways in the CNS [12,54,55,56]. Taken together, the results of the current study, associated with our previous results showing that only WSB/EiJ mice were protected from hypothyroidism-induced neuroinflammation [15], suggest that high metabolic flexibility could be a protective factor against neuroinflammation-induced neurodegenerative diseases.

## Figures and Tables

**Figure 1 ijms-25-10792-f001:**
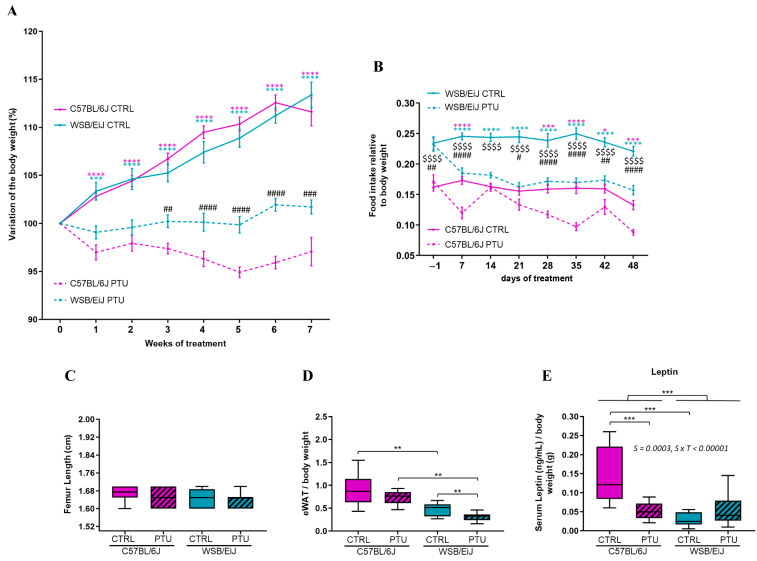
PTU treatment effects on metabolism including body weight, food intake and body fat mass. (**A**) Body weight variation upon weeks of treatment expressed as a percentage of the starting body weight (Week 0) for each group (% each week BW/Initial BW before the treatment; *n* = 13–14 per group). (**B**) Food intake measurements (relative to BW) during the 7 weeks of treatment (*n* = 13–15 per group). Food intake data were collected starting from 24 h before the treatment (−1). (**A**,**B**) The statistical differences between treatment and time were assessed by two-way repeated measures ANOVA followed by Tukey’s multiple-comparisons test. Significant differences were indicated by different symbols to account for group differences at each week (blue and pink *: WSB/EIJ CTRL vs. PTU and C57BL/6J CTRL vs. PTU, respectively; $: C57BL/6J CTRL vs. WSB/EIJ CTRL; #: C57BL/6J PTU vs. WSB/EIJ PTU; *,#, *p* ≤ 0.05; ##, *p* ≤ 0.01; ###, *p* ≤ 0.001; ####, $$$$, *p* ≤ 0.0001). Values are mean ± SEM. (**C**) Effect of hypothyroidism on bone growth. Femur length measurements were not affected by PTU treatment in C57BL/6J and WSB/EiJ mice (*n* = 8–12 per group; non-parametric two-way ANOVA with permutations test, *p* > 0.05). (**D**) Epididymal white adipose tissue (eWAT) weights measured at the end of the treatment (relative to final BW; *n* = 12–15 per group; non-parametric one-way ANOVA with permutations test). (**E**) Circulating leptin (in ng/mL; *n* = 9–13 per group) Boxplot represents median values and min–max whiskers. Non-parametric two-way ANOVA with permutations test were performed for (**E**) data (see Appendix A). Statistically significant effects with the respective *p*-values are indicated on the graph (S: strain, S × T: strain × treatment interaction). Post hoc tests results are indicated on the graphs (**, *p* ≤ 0.01; ***, *p* ≤ 0.001; ****, *p* ≤ 0.0001).

**Figure 2 ijms-25-10792-f002:**
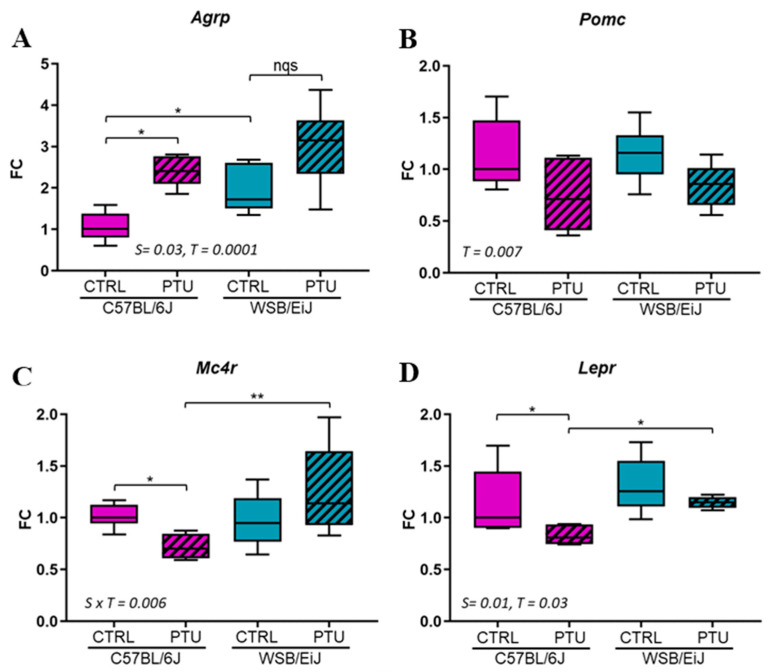
PTU treatment effect on hypothalamic control of energy balance. Expression of appetite-stimulating (orexigenic) neuropeptide agouti-related peptide (*Agrp*) (**A**), appetite-suppressing (anorexigenic) neuropeptides pro-opiomelanocortin (*Pomc*) (**B**), melanocortin 4 receptor (*Mc4r)* (**C**) and leptin receptor (*Lepr*) (**D**) in response to 7weeks of PTU treatment. Data are represented as relative fold-change expression (FC). Boxplots represent median values and min–max whiskers. Non-parametric one-way ANOVA with permutations post hoc tests results are indicated on the graph (*n* = 5–6 per group; *, *p* ≤ 0.05; **, *p* ≤ 0.01; nqs: not quite significant). Non-parametric two-way ANOVA with permutations test: statistically significant effects with the respective *p*-values are indicated on the graph (S: strain, T: treatment, S × T: strain × treatment interaction). When interaction *p*-value is significant, post hoc test results are indicated on the figure.

**Figure 3 ijms-25-10792-f003:**
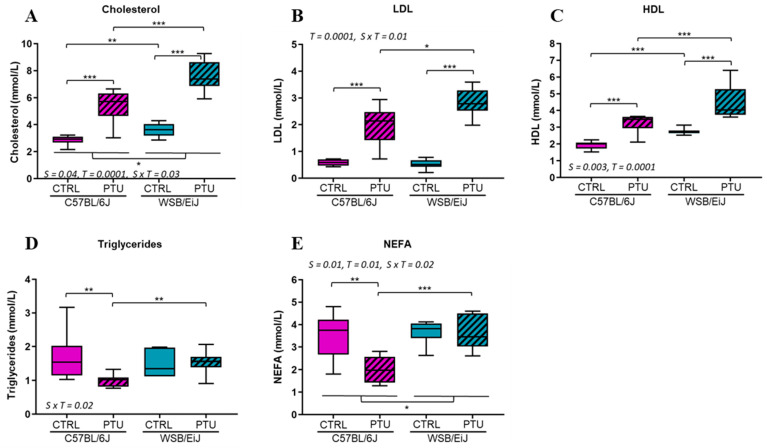
Circulating lipid patterns are differentially altered between mouse strains in response to hypothyroidism. (**A**) Cholesterol (in mmol/L), (**B**) low-density lipoprotein fatty acid (LDL in mmol/L) and (**C**) high-density lipoprotein (HDL in mmol/L) levels were increased in both mouse *p*-values as indicated on the graph (S: strain, T: treatment, S × T: strain × treatment interaction). Strains after PTU treatment. (**D**) Triglycerides (in mmol/L) and (**E**) non-esterified fatty acid (NEFA in mmol/L) levels were decreased in hypothyroid C57BL/6J mice whereas they remained unchanged in WSB/EiJ mice. Boxplot represents median values and min–max whiskers. Non-parametric one-way ANOVA with permutations post hoc tests results are indicated on the graph (*n* = 7–8 per group; *, *p* ≤ 0.05; **, *p* ≤ 0.01; ***, *p* ≤ 0.001). Non-parametric two-way ANOVA with permutations test: statistically significant effects with the respective *p*-values are indicated on the graph (S: strain, T: treatment, S × T: strain × treatment interaction). When interaction *p*-value is significant, post hoc test results are indicated on the figure.

**Figure 4 ijms-25-10792-f004:**
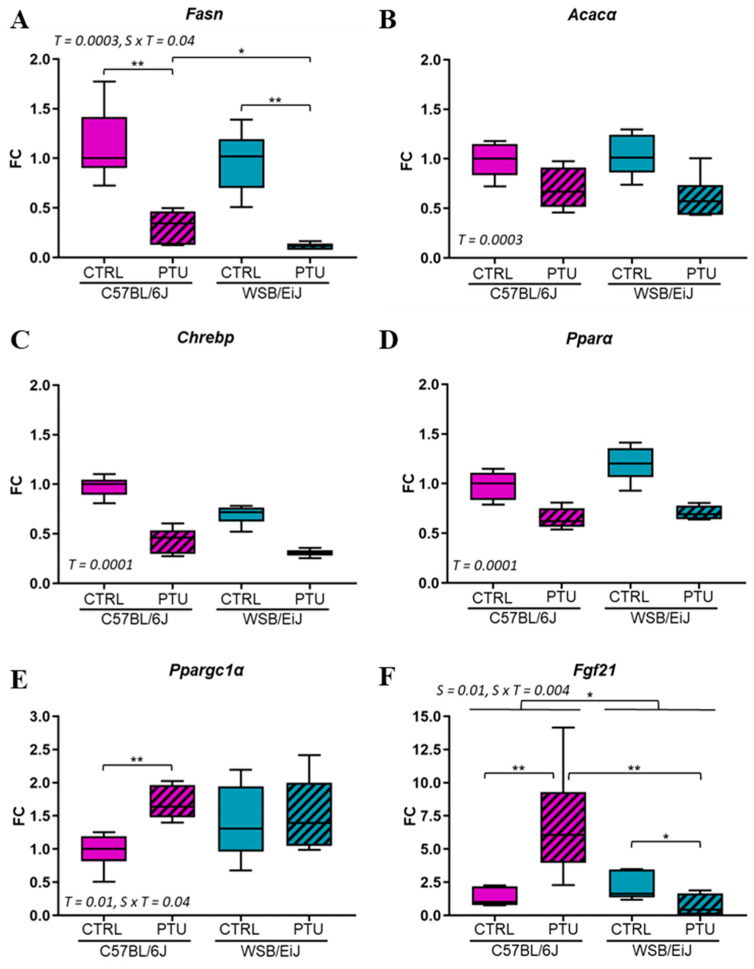
Hepatic lipid metabolism is reduced in both strains in response to hypothyroidism. Hepatic expression of lipogenesis key genes *Fasn* (**A**) and *Acacα* (**B**) involved in the fatty acid and TGs synthesis (referred as de novo lipogenesis (DNL)) were downregulated in both strains after PTU treatment as well as *Chrebp* expression (**C**), the DNL-regulated transcription factor. Expression of *Pparα* (**D**), *Ppargc1α* (**E**) and *Fgf21* (**F**) genes involved in lipid metabolism regulation and fatty acid oxidation were differentially regulated between strains. Data are represented as relative fold-change expression (FC). Boxplot represents median values and min–max whiskers. Non-parametric two-way ANOVA with permutations tests: statistically significant effects with the respective *p*-values are indicated on the graph (S: strain, T: treatment, S × T: strain × treatment interaction). When the interaction *p*-value is significant, post hoc test results are indicated on the figure (*n* = 5–6 per group; *, *p* ≤ 0.05; **, *p* ≤ 0.01).

**Figure 5 ijms-25-10792-f005:**
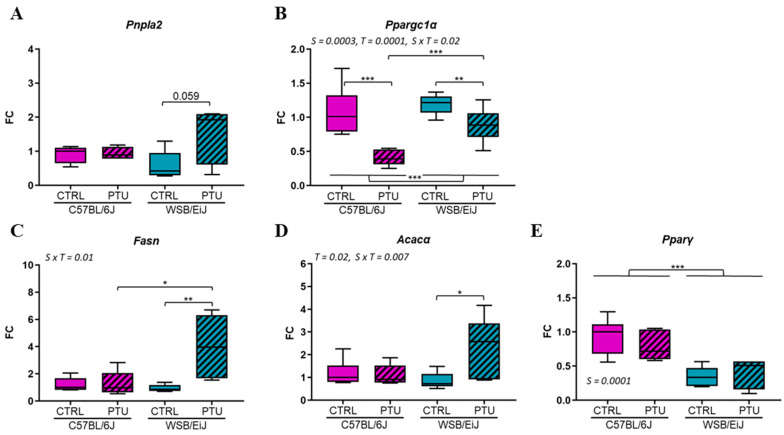
Adipose lipid metabolism is globally enhanced in WSB/EiJ mice and unchanged in C57BL/6J mice under PTU treatment. (**A**) Expression of *Pnpla2* transcript, catalyzing TGs into fatty acids (referred as lipolysis) seemed to be increased only in WSB/EiJ mice but did not reach significance. (**B**) Expression of *Ppargc1α* gene, involved in fatty acid oxidation, was downregulated between strains. Expression in the eWAT of lipogenesis key genes *Fasn* (**C**) and *Acacα* (**D**) involved in the fatty acid and TGs synthesis was upregulated only in WSB/EiJ mice after PTU treatment. (**E**) Expression of the DNL-regulated transcription factor *Pparγ* gene was different between strains. Data are represented as relative fold-change expression (FC). Boxplot represents median values with min–max whiskers. Non-parametric two-way ANOVA with permutations tests: statistically significant effects with the respective *p*-values are indicated on the graph (S: strain, T: treatment, S × T: strain × treatment interaction). When interaction *p*-value is significant, post hoc test results are indicated on the figure (*n* = 5–6 per group; *, *p* ≤ 0.05; **, *p* ≤ 0.01; ***, *p* ≤ 0.001).

**Figure 6 ijms-25-10792-f006:**
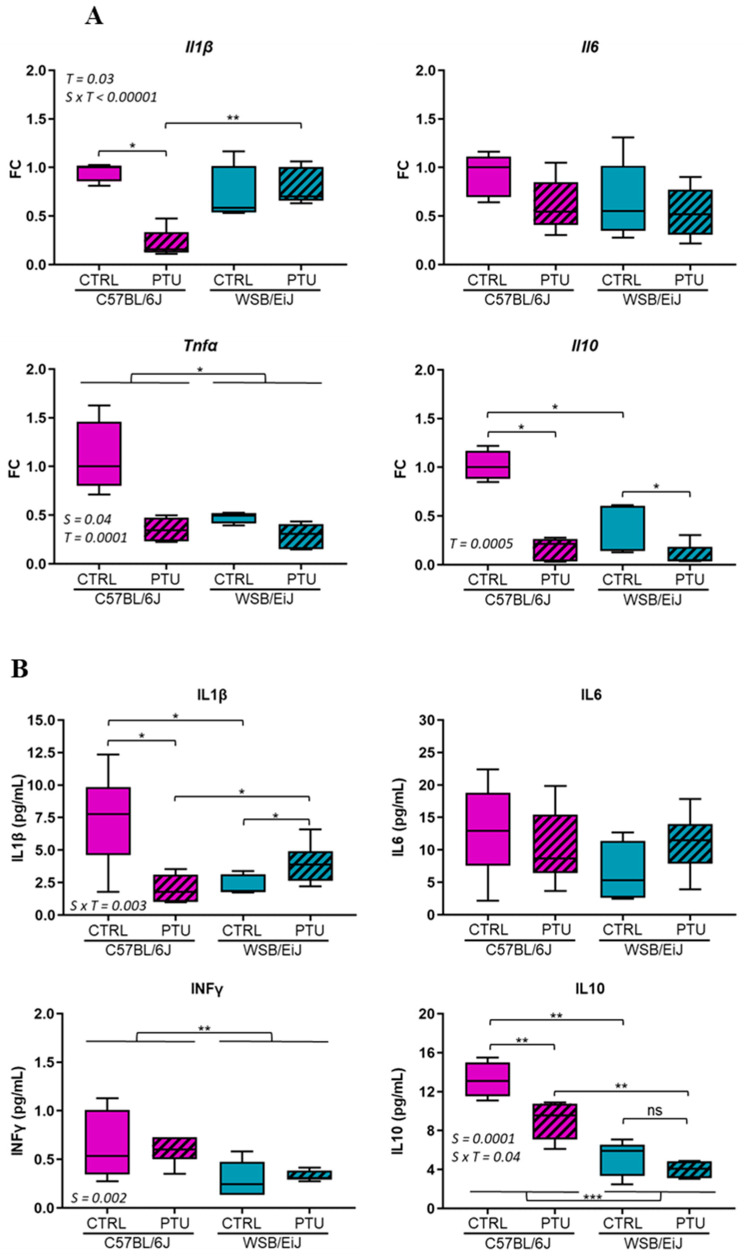
Effect of PTU treatment on inflammatory cytokines. (**A**) Expression of pro-inflammatory *Il1B*, *Il6*, *Tnfα*, and anti-inflammatory *Il10* cytokine genes in the eWAT were overall downregulated in hypothyroid C57BL/6J mice, whereas they remained unchanged in WSB/EiJ mice. Data are represented as relative fold-change expression (FC). (**B**) Circulating inflammatory cytokines measured by ELISA: IL10 (in pg/mL), IL1β (in pg/mL), IFNγ (in pg/mL), and IL6 (in pg/mL) levels were overall decreased in hypothyroid C57BL/6J mice, whereas they remained unchanged in WSB/EiJ mice. Boxplot represents median values with min–max whiskers. Non-parametric one-way ANOVA with permutations post hoc tests results are indicated on the graph (*n* = 5–6 per group; *, *p* ≤ 0.05; **, *p* ≤ 0.01; ***, *p* ≤ 0.001; *ns*, not significant). Non-parametric two-way ANOVA with permutations tests: statistically significant effects with the respective *p*-values are indicated on the graph (S: strain, T: treatment, S × T: strain × treatment interaction). When interaction *p*-value is significant, post hoc test results are indicated on the figure.

**Figure 7 ijms-25-10792-f007:**
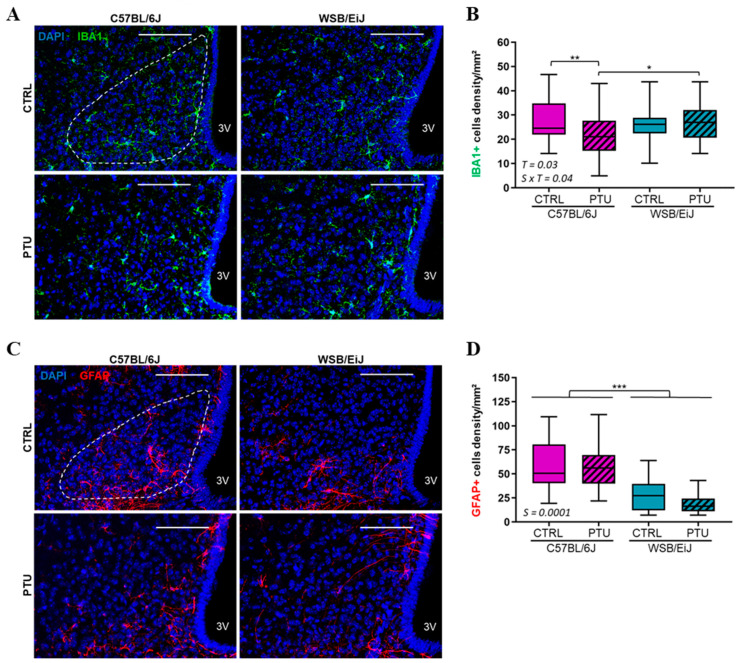
Effect of hypothyroidism on glial cells density in the hypothalamic arcuate nucleus (ARC) of mouse strains. (**A**) Representative confocal images of IBA1+ microglia (green), in the ARC (white ROI) of euthyroid (CTRL) and hypothyroid (PTU) C57BL/6J (left panel) and WSB/EiJ (right panel) mice. Cells nuclei are stained with DAPI (in blue). (**B**) Quantitative analysis revealed a decrease in microglia density in the ARC of C57BL/6J mice, whereas it was unchanged in WSB/EiJ mice in response to PTU treatment. (**C**) Representative confocal images of GFAP+ astrocytes (red), in the ARC of euthyroid (CTRL) and hypothyroid (PTU) C57BL/6J (left panel) and WSB/EiJ (right panel) mice. Cells nuclei are stained with DAPI (in blue). (**D**) Density of GFAP+ astrocytes remained unchanged in the ARC of both mouse strains after PTU treatment, and a strain effect revealed lower astrocyte density in the ARC of WSB/EiJ compared to C57BL/6J mice. Boxplot represents median values with min–max whiskers. Strain effect result is indicated on the graph (*n* = 4 mice per group, *n* = 2–6 sections per mouse and *n* = 2 ROI per section). Non-parametric two-way ANOVA with permutations tests: statistically significant effects with the respective *p*-values are indicated on the graph (S: strain, T: treatment, S × T: strain × treatment interaction). When the interaction *p*-value is significant, post hoc test results are indicated on the figure. *, *p* ≤ 0.05; **, *p* ≤ 0.01; ***, *p* ≤ 0.001. Scale bars = 100 μm.

**Figure 8 ijms-25-10792-f008:**
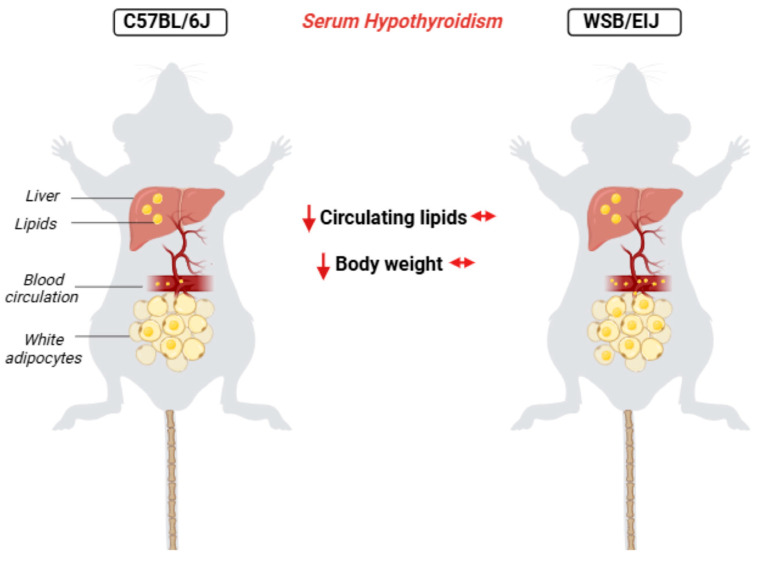
Metabolic flexibility allows for adaptation to circulating hypothyroidism: Circulating hypothyroidism induces different consequences on peripheral lipid metabolism in C57BL/6J and WSB/EiJ mouse strains. In C57BL/6J mice (**left**) and WSB/EiJ mice (**right**), circulating hypothyroidism induces an alteration in hepatic lipid metabolism, resulting in a decreased lipid synthesis in the liver. However, compensatory mechanisms (mobilization of WAT lipid stores) occur only in WSB/EiJ mice to maintain their circulating levels of NEFA and TG despite hypothyroidism, and therefore prevent hypothyroidism-induced weight loss, unlike in C57BL/6J mice. These results highlight the importance of metabolic flexibility in order to adapt to TH level disruption. *This figure was created with BioRender.com.*

## Data Availability

The raw data supporting the conclusions of this article will be made available by the authors on request.

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
