# Peer review of "The Downregulation of the Liver Lipid Metabolism Induced by Hypothyroidism in Male Mice: Metabolic Flexibility Favors Compensatory Mechanisms in White Adipose Tissue"

_ijms, 2024, doi:10.3390/ijms251910792_

Round 1

Reviewer 1 Report

Comments and Suggestions for Authors

The authors compared the response to induced hypothyroidism in two mouse strains, the wild-derived WSB/EiJ mouse strain characterized by a diet-in-duced obesity (DIO) resistance due to its high metabolic flexibility phenotype and the C57BL/6J mice, prone to DIO. Our results show that PTU-induced hypothyroidism led to metabolic deregulations, particularly a reduction in hepatic lipid synthesis in both strains. Furthermore, contrary to the C57BL/6J mice, the WSB/EiJ mice were resistant to the metabolic dysregulations induced by hypothyroidism, mainly through an enhanced lipid metabolism in adipose tissue. Indeed, WSB/EiJ mice compensated the decrease in hepatic lipid synthesis by a mobilization of lipid reserves from white adipose tissue. Gene expression analysis revealed that hypothyroidism stimulates the hypothalamic orexigenic circuit in both strains, but unchanged Mc4r and LepR expression in hypothyroid WSB/EiJ mice strain, which reflect their adaptability to maintain their body weight, contrary to C57BL/6J mice. Thus, our results show that WSB/EiJ mice displayed a resistance to metabolic dysregulations induced by hypothyroidism, by compensatory mechanisms. This highlights the importance of metabolic flexibility in the ability to adapt to disturbed circulating TH levels. Its interesting for readers.

However, the paper does not have an integrated structure and the meanings of some sentences are not clear; the language is not good to be published. As a result, the paper could not be accepted in the journal in its present form and some improvements are required to make before published.

Abstract

1. Generally, the paper is written in the third-person narrative form, so, the our objective and our result should be replaced by the aim of the paper and the result”.

Introduction

1. line 40, (review in [1]) should be replaced an [1], same as line 44 (review in [1, 2]) and line 51.

2. The paper should be described using the passive voice, so, the sentence in line 90 and line 97 need to be rewritten.

Results

1. Line 108 and 111 are not right form, please rewrite.

2. Line 127, 130, 132 the p should be P. the same as below.

Discussion

1. In line 443, confirm should be confirmed.

2. Line 459 should be rewritten.

Conclusion

Conclusion is different from discussion, so, in line 598 and line 602, no references could be appeared in this section.

Comments on the Quality of English Language

The paper does not have an integrated structure and the meanings of some sentences are not clear; the language is not good to be published. As a result, the paper could not be accepted in the journal in its present form and some improvements are required to make before published.

Author Response

Please, see attachment for responses to reviewer 1's comments

Reviewer 2 Report

Comments and Suggestions for Authors

1. The thyroid hormone action can be sexually dimorphic. Please revise the title and the abstract to clarify that the study only uses male mice.

2. Please use dot plots (instead of bar charts) to show each data point.

3. Please provide details on how to determine which portion is the hypothalamus for tissue collection.

4. Fig. 1B. The X-axis should be labeled in days, not weeks.

5. Line 404. There should be a period between "reduced" and "Therefore."

6. The source of antibodies should be provided.

Author Response

Please, see the attachment for responses to reviewer 2's comments

Reviewer 3 Report

Comments and Suggestions for Authors

In their paper entitled “Down regulation of the liver lipid metabolism induced by hypothyroidism in mice: metabolic flexibility favors compensatory mechanisms in white adipose tissue”, the Authors report the results of a study aimed at comparing the response to induced hypothyroidism of two different mouse strains (C57BL/6J and WSB/EiJ), one of which (WSB/EiJ) was already known for its metabolic flexibility. The results show that hypothyroidism induces a decrease of lipid synthesis in the liver, in both strains; interestingly, however, only the WSB/EiJ strain is able to respond by increasing lipid mobilization from the white adipose tissue. Since metabolic unbalance induces hypothalamic inflammation, the Authors also analyzed this aspect in their experiments. On the basis of the presented data, the Authors conclude that high metabolic flexibility could be a protective factor in many alterations of the metabolic conditions, and also in neuroinflammation-induced neurodegenerative diseases.

The paper is of interest and suitable for International Journal of Molecular Sciences.

I have three main suggestions for the Authors:

1.In most books and papers discussing the effects of thyroid hormones, hypothyroidism is often considered a cause of body weight gain, because of less energy expenditure, and probably, as the Authors of the present paper also report, because of modifications at the level of hypothalamus, with an increase of orexigenic signals. As in the present paper, PTU-induced hypothyroidism induces body weight decrease, I think that these interesting findings, probably also related to the mice age, might be commented also in relation with what is normally reported (a brief comment on the general effects of hypothyroidism is already present at lines 59-61 of the paper);

2. It is now clear that hypothyroidism can also derive from mutations in the thyroid hormone nuclear receptors, and, in particular, from mutations in the hormone-binding domain. This condition, known as thyroid hormone resistance, depending on the class of receptors affected, can be characterized by symptoms of hypothyroidism, in the presence of almost normal or even high concentrations of blood thyroid hormones. I think that a brief information on this possibility, in Introduction, might be of interest for the Readers.

3. In my opinion, the supplementary figures and tables might be included in the main text.

Minor comments:

Abstract, lines 21 and 27 – PTU, Mc4r, and LepR: please give the complete meaning and not only the abbreviations;

Introduction, line 45 – please provide the complete names of these neurons (NPY/AGRP, and POMC/CART);

Author Response

Please, see the attachment for responses to reviewer 3's comments
